# Inflammatory markers for improved recurrent UTI diagnosis in postmenopausal women

Tahmineh Ebrahimzadeh[1,*], Ujjaini Basu[1,*] ⬤, Kevin C Lutz[2] ⬤, Jashkaran Gadhvi[1] ⬤, Jessica V Komarovsky[1] ⬤, Qiwei Li[2] ⬤, Philippe E Zimmern[3] ⬤, Nicole J De Nisco[1,3] ⬤

Recurrent urinary tract infection (rUTI) severely impacts postmenopausal women. The lack of rapid and accurate diagnostic tools is a major obstacle in rUTI management as current gold standard methods have >24-h diagnostic windows. Work in animal models and limited human cohorts have identified robust inflammatory responses activated during UTI. Consequently, urinary inflammatory cytokines secreted during UTI may function as diagnostic biomarkers. This study aimed to identify urinary cytokines that could accurately diagnose UTI in a controlled cohort of postmenopausal women. Women passing study exclusion criteria were classified into no UTI and active rUTI groups, and urinary cytokine levels were measured by immunoassay. Pro-inflammatory cytokines IL-8, IL-18, IL-1$\beta$, and monocyte chemoattractant protein-1 were significantly elevated in the active rUTI group, and anti-inflammatory cytokines IL-13 and IL-4 were elevated in women without UTI. We evaluated cytokine diagnostic performance and found that an IL-8, prostaglandin E2, and IL-13 multivariable model had the lowest misclassification rate and highest sensitivity. Our data identify urinary IL-8, prostaglandin E2, and IL-13 as candidate biomarkers that may be useful in the development of immunoassay-based UTI diagnostics.

## Introduction

Urinary tract infection (UTI) poses a significant healthcare burden worldwide, affecting more than 400 million people per year (Yang et al, 2022). UTI disproportionately affects women and older adults (Rowe & Juthani-Mehta, 2013). UTI management heavily relies on antibiotic therapy (Colgan & Williams, 2011). However, the increasing prevalence of antimicrobial resistance complicates treatment and results in higher hospitalization rates (Hooton et al, 2004). About 20–30% of premenopausal women and ~50% of postmenopausal women with UTI will experience another UTI

within 6 mo and develop recurrent UTI (rUTI) (Foxman, 2002; Glover et al, 2014).

Accurate diagnosis of UTI remains a clinical challenge (Claeys et al, 2019). Initial UTI diagnosis is based on symptoms, physical examination, and urinalysis (Meister et al, 2013). Urine culture (UC) is the gold standard to confirm UTI diagnosis but has a diagnostic window of 48–72 h (Davenport et al, 2017). Consequently, broad-spectrum antibiotics are often prescribed for early empiric treatment of UTI (Claeys et al, 2019). Diagnosis of UTI is particularly challenging in the elderly because mental and physical comorbidities make differentiation of UTI from other urological disorders based on symptoms alone difficult (Mambatta et al, 2015; Cortes-Penfield et al, 2017). The most commonly used point-of-care UTI diagnostic, the urine dipstick, uses leukocyte esterase (LE) as an indicator of pyuria and nitrate (Ni) as an indicator of bacteriuria (Lammers et al, 2001). However, the urine dipstick suffers from poor specificity and a high false-positive rate (Mambatta et al, 2015).

Inflammatory cytokines secreted as part of the immune response to infection have the potential to be useful diagnostic biomarkers (Ko et al, 1993; Rao et al, 2001; Frimpong et al, 2022). There is an established association between local inflammation and UTI severity and recurrence in humans and animal models (Czaja et al, 2009; Sivick et al, 2010; Ingersoll & Albert, 2013). Inflammation contributes to the development of the clinical symptoms of dysuria, urgency, and frequency during UTI (Brierley et al, 2020). Studies in mouse models suggest that early immune checkpoint activation during infection with uropathogenic *Escherichia coli* (UPEC), the most common cause of UTI, determines host susceptibility to rUTI (Hannan et al, 2012; O'Brien et al, 2016). Significant elevation of urinary levels of keratinocyte-derived protein chemokine (KC), IL-6, and G-CSF is predictive of chronic cystitis in mouse models (Hannan et al, 2010; Yu et al, 2019). KC and MIP-2, which are functional orthologues of human IL-8 and are involved in neutrophil trafficking, are among the first cytokines detected during UTI in mice (Hang et al, 1999). Immune profiling of urinary cytokines in adults with UPEC UTI detected IL-

[1]Department of Biological Sciences, University of Texas at Dallas, Dallas, TX, USA   [2]Department of Mathematics, University of Texas at Dallas, Dallas, TX, USA   [3]Department of Urology, University of Texas Southwestern Medical Center, Dallas, TX, USA

Correspondence: nicole.denisco@utdallas.edu
*Tahmineh Ebrahimzadeh and Ujjaini Basu contributed equally to this work

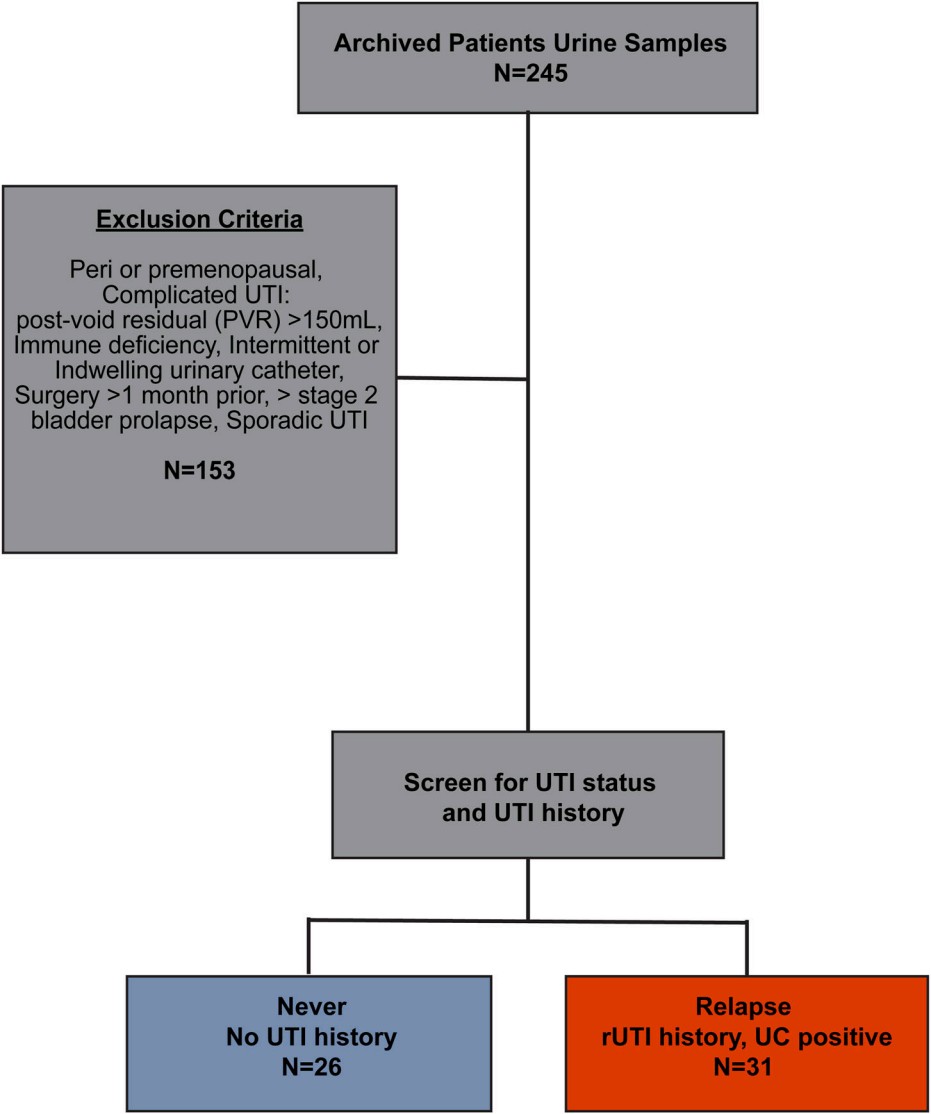

**Figure 1. Study flowchart, cohort design, and selection criteria.**
Urine samples from 245 patients were screened using study exclusion criteria and cohort design. Patients who passed the exclusion criteria and either had no urinary tract infection history (never, N = 26) or had recurrent urinary tract infection history and a current urinary tract infection (relapse, N = 31) were included in the study.

1$\beta$, IL-6, IL-17A, CCL2, monocyte chemoattractant protein-1 (MCP-1), TNF-$\alpha$, and IFN-$\gamma$ (Ambite et al, 2016; Sundac et al, 2016; Armbruster et al, 2018). Interestingly, one study found urinary IL-8 to be elevated in 92% of infected patients independent of the species of causative uropathogen (Oregioni et al, 2005).

Despite the role of inflammation and innate immunity in UTI outcome, few immune marker-based diagnostics for UTI have been developed or implemented. Studies to identify inflammatory biomarkers for UTI in human populations are limited, especially in postmenopausal women (Franz & Horl, 1999; Czaja et al, 2009; Glaser & Schaeffer, 2015; Soric Hosman et al, 2022). There has been no published comprehensive study evaluating the sensitivity and selectivity of multiple urinary cytokines for UTI diagnosis in a controlled population of postmenopausal women. Therefore, the aim of this study was to investigate urinary cytokines as diagnostic markers for rUTI in postmenopausal women.

## Results

### Cohort design and patient characteristics

We conducted a cross-sectional study to identify urinary biomarkers for the detection of rUTI in postmenopausal women. 57 out of 245 (23%) enrolled patients passed the exclusion criteria and fell into either the never (no UTI history, no UTI symptoms, N = 26) or relapse (rUTI history, current UTI symptoms, positive UC, N = 31) group (Fig 1). Over 82% of the participants were aged > 65 years (Table 1). The median age of women in the relapse group (74 yr) was significantly higher than women in the never group (69.5 yr, $P$ = 0.0263) (Table 1). Median body mass index (BMI) in the relapse group was also significantly higher ($P$ = 0.0097). There was no significant difference between the number of diabetic (AODM) patients in the relapse versus the never group ($P$ = 0.075). No women in the never group reported antibiotic use one week before sample

**Table 1. Subject demographics and clinical characteristics.**

| Patient characteristics | Never | Relapse | P-value |
|---|---|---|---|
| Total | N = 26 | N = 31 | |
| Median age (years) | 69.5 | 74 | P = 0.026 |
| <65 | N = 7 (26.9%) | N = 3 (9.7%) | |
| ≥65 | N = 19 (73.1%) | N = 28 (90.3%) | |
| Median BMI (kg/m$^2$) | 26.05 | 27.9 | P = 0.011 |
| 18.5–24.9 | N = 9 (34.6%) | N = 5 (16.1%) | |
| 25–29.9 | N = 14 (53.9%) | N = 15 (48.4%) | |
| 30–34.9 | N = 2 (7.7%) | N = 7 (22.6%) | |
| >35 | N = 1 (3.8%) | N = 4 (12.9%) | |
| Median pH | 6 | 5.35 | P = 0.078 |
| ≤6 | N = 17 (65.4%) | N = 24 (77.4%) | |
| >6 | N = 9 (34.6%) | N = 7 (22.6%) | |
| Smoker | | | |
| Active | N = 1 (3.8%) | N = 0 (0%) | |
| Former hx | N = 10 (38.5%) | N = 8 (25.8%) | |
| Diabetic (AODM) | N = 1 (3.8%) | N = 6 (19.4%) | P = 0.075 |
| CFT | N = 0 (0%) | N = 19 (61.3%) | |
| Antibiotics | N = 0 (0%) | N = 10 (32.3%) | |

BMI, body mass index; AODM, adult-onset diabetes mellitus; CFT, cystoscopy with fulguration of trigonitis. The *t* test was used to calculate the *P*-value for data that were normally distributed (age). The Mann–Whitney *U* test was used to calculate *P*-value for data that were not normally distributed (BMI and pH). The chi-square test was used for categorical data (diabetic).

collection, whereas 10 women in the relapse group were currently taking antibiotics. Nineteen relapse patients had a prior history of cystoscopy with fulguration of trigonitis (CFT) for advanced rUTI management (Crivelli et al, 2019).

## Urinary symptoms and urinalysis (UA) of the relapse group

Patients in the relapse group presented with symptoms suggestive of UTI during physical examination. Twenty-nine percent of the patients experienced dysuria, and 68% and 52% reported frequency and urgency, respectively (Fig 2A). Other symptoms reported included back pain, voiding difficulty, and cloudy urine. Urinalysis (UA) via a urine dipstick was performed on all relapse urine samples. 61% of relapse patients' urine samples were positive for Ni, and 80% were positive for LE (Fig 2B). In total, 87% of patients were positive for either Ni or leukocyte esterase (LE) and 55% were positive for both Ni and LE. Blood (>trace) was detected in the urine of 68% of the relapse patients (Fig 2A and B). UPEC infection was observed with the highest frequency (48.4%) in relapse patients (Fig 2C), followed by *Klebsiella pneumoniae* (16.1%) and *Enterococcus faecalis* (9.7%). Less frequently observed pathogens were *Klebsiella oxytoca* (6.5%), *Streptococcus agalactiae* (3.2%), *Pseudomonas aeruginosa* (3.2%), and *Klebsiella aerogenes* (3.2%) (Fig 2C). Only the major dominant species in each urine culture is depicted in Fig 2C, but complete clinical urine culture results are available in Table S1.

## Urinary cytokines are differentially abundant in postmenopausal women with active rUTI

To identify differentially abundant cytokines in women with active rUTI (relapse) versus no UTI (never), urinary levels of 20 cytokines were measured by bead-based immunoassay in a subset of the never (N = 23) and relapse groups (N = 23) (Fig S1). We found that the median concentrations of four inflammatory cytokines, IL-1β, MCP-1, IL-8, and IL-18, were higher in the relapse compared with the never group (Fig 3A–D). The median concentrations of IL-1β, MCP-1, IL-8, and IL-18 were, respectively, 23.4, 7.73, 108.6, and 4.78 times higher in the relapse versus the never group (Table 2). Because urinary cytokine levels may be affected by urine concentration, they are often normalized to urinary creatinine (Cr). We found that both the raw and Cr-normalized urinary concentrations of IL-1β, MCP-1, IL-8, and IL-18 were significantly elevated in the relapse compared with the never group (Fig 3A–D, Table 2).

Two anti-inflammatory cytokines, IL-4 and IL-13 were differentially abundant between relapse and never groups. Both raw and Cr-normalized urinary IL-13 levels were significantly elevated in the never compared with the relapse group (Fig 3E). However, only Cr-normalized IL-4 was slightly higher in the never group (Fig 3F). It should be noted that although IL-13 and IL-4 were present in some never patients, for most patients, the concentration was below the limit of detection (LoD) of the assay, and therefore, the median raw concentration of IL-4 and IL-13 remained at the LoD for both groups (Table 2).

We then performed clustering and correlation analysis to determine if any of the urinary cytokines co-occur. Hierarchical clustering revealed that the measured cytokines fell into six clusters (Fig S2A), with the significantly enriched inflammatory cytokines IL-1β, MCP-1, IL-8, and IL-18 and previously measured prostaglandin E2 (PGE2) all falling into the cluster (Ebrahimzadeh et al, 2021). Furthermore, urinary concentrations of these six inflammatory cytokines were all significantly and strongly correlated with *P* > 0.5 and *P* < 0.05 for all pairwise associations (Fig S2B).

## IL-8, PGE2, and IL-13 are strongly predictive of rUTI status in postmenopausal women

Previous work demonstrated that urinary PGE2 was significantly higher in postmenopausal women with rUTI history and current UTI (relapse) compared with postmenopausal women without UTI history (never) and was highly predictive of rUTI status (Ebrahimzadeh et al, 2021). We hypothesized that other urinary cytokines may improve the performance of the PGE2 diagnostic model. We selected six differentially abundant cytokines between the never and relapse cohorts, IL-1β, MCP-1, IL-18, IL-8, IL-13, and IL-4, for logistic regression analysis. Urinary PGE2, which was measured in these samples in a previous study (Ebrahimzadeh et al, 2021), was also tested to directly compare single and multivariate models.

First, cytokines were tested individually to determine their association with rUTI status in a single-variable model (Fig 4A). IL-8 was the best-performing model for discrimination of patients based on the current rUTI status area under the curve (AUC = 0.885). The sensitivity and specificity of the IL-8 model were 0.739 and 0.957,

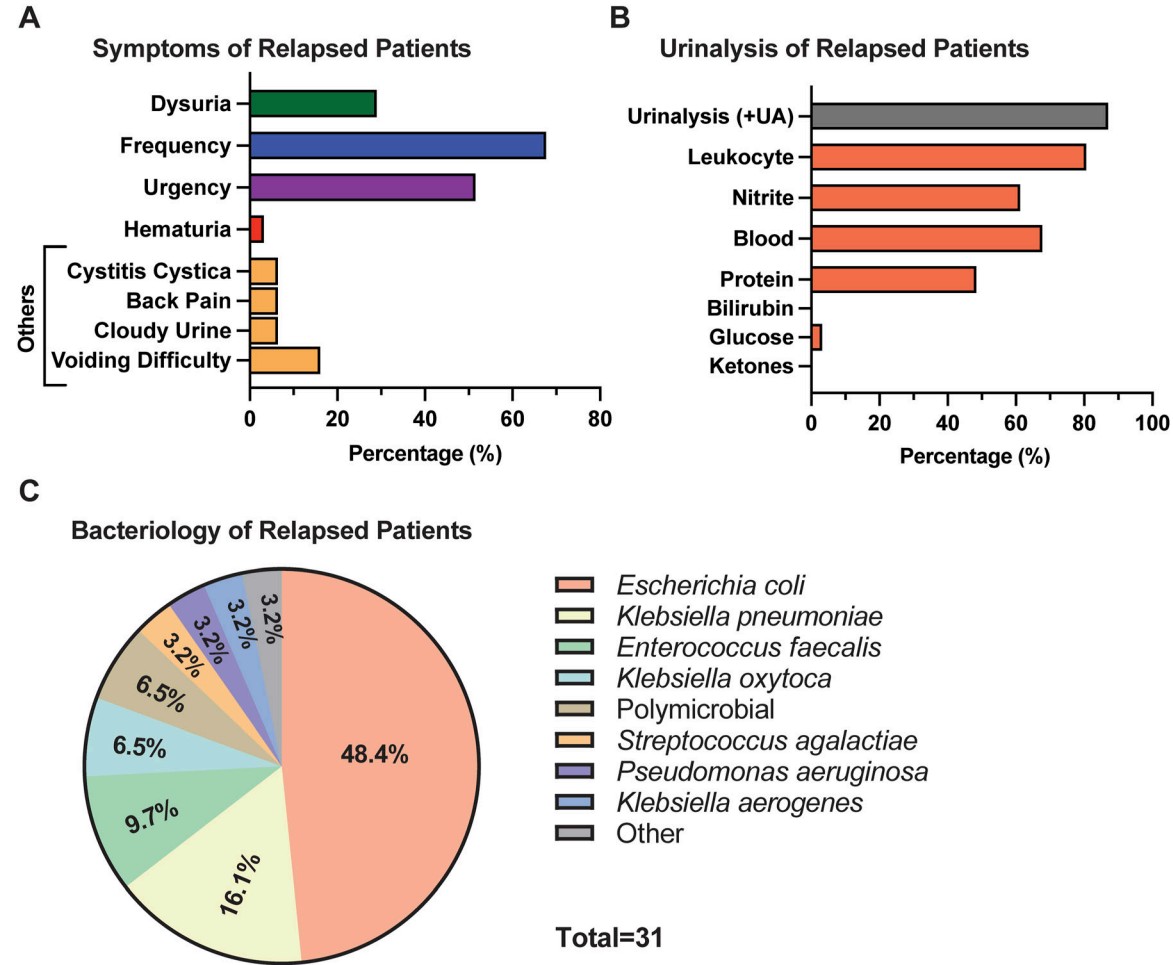

**Figure 2. Reported symptoms and urinalysis results.**
**(A)** Bar graph represents the percentage of patients with symptoms associated with urinary tract infection in the relapse cohort. **(B)** Bar graph depicts urinalysis results of the relapse patients. Patients with LE or Ni > +1 were considered UA-positive. The presence of blood and protein (>trace) are reported. **(C)** Pie chart showing the major uropathogens detected in the urine of the relapsed patient group by clinical urine culture. Full bacteriology data are found in Table S1.

respectively. Moreover, IL-8 was the best fitted model (pseudo-$R^2$ = 0.476) with the lowest misclassification rate (0.152) (Table 3). IL-18 was the second best-performing model in terms of AUC; however, PGE2 outperformed IL-18 in all other predictive metrics (Table 3). Interestingly, neither IL-8 nor PGE2 were associated with specific bacterial uropathogenic species (Fig S3A and B). A history of fulguration also did not seem to influence the IL-8 or PGE2 levels (Fig S4A and B). We also evaluated the contribution of clinical variables like age, BMI, and urine pH to model accuracy using a bagged logistic regression model and found that the top differentially enriched cytokines contributed most to model accuracy. Removal of BMI and age resulted in less than a 5% decrease in mean accuracy and therefore do not contribute significantly to the diagnostic model (Fig S4C).

Next, we analyzed the association of the urinary cytokines with rUTI in a two-variable model. A total of 21 models passed the cutoff for statistical significance (Table S2). Although PGE2 and IL-13 together had the highest AUC (0.919), the PGE2 and IL-8 model had a slightly lower AUC (0.904) but a better misclassification rate (0.152),

sensitivity (0.826), specificity (0.870), and pseudo-$R^2$ (0.601) (Fig 5B) (Table S2). This discrepancy can be explained by the fact that AUC does not account for misclassification rate and model fit. Therefore, we prioritized misclassification rate over AUC as long as the AUC remained sufficiently high (0.9–1) in the selected model. Based on these metrics, the PGE2 and IL-8 model best discriminated between never and relapse groups in this cohort of postmenopausal women (Table S2).

Lastly, logistic regression analysis of three-variable models found a total of 35 statistically significant models (Table S3). The PGE2, IL-18, and IL-13 model had the highest AUC (0.930); however, the PGE2, IL-8, and IL-13 model had an AUC of 0.905 and outperformed all other three-variable models in terms of misclassification rate (0.109), sensitivity (0.913), and pseudo-$R^2$ (0.797) (Fig 4C, Table S3). The top-performing single-, two-, and three-variable models are compared in Fig 4D and Table 4. Our data indicate that overall, the PGE2, IL-8, IL-13 model had excellent sensitivity and specificity with the lowest misclassification rate for rUTI diagnosis in postmenopausal women.

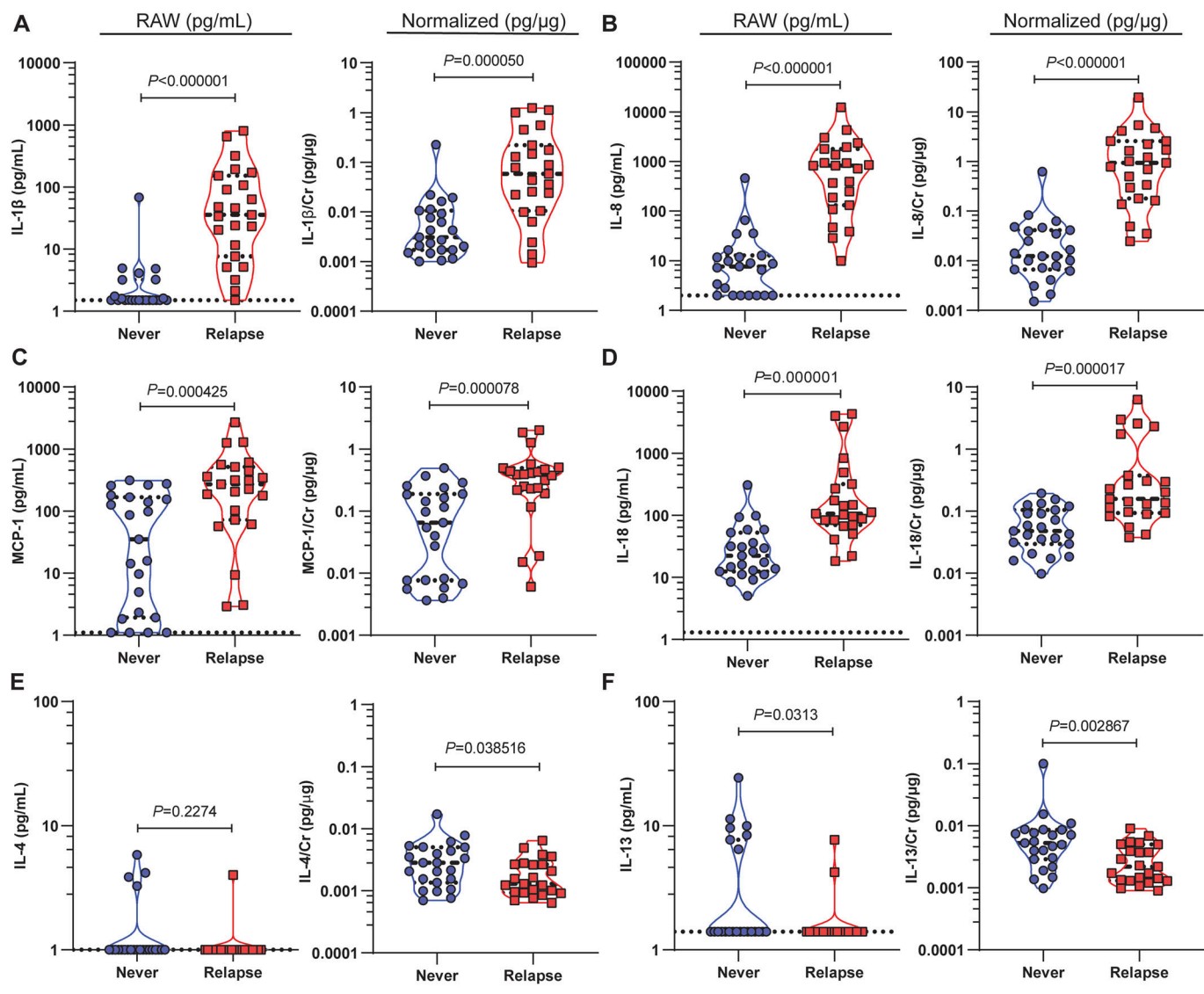

**Figure 3.  Differential elevation of pro- and anti-inflammatory cytokines between never and relapse groups.**
**(A, B, C, D)** Raw and creatinine (Cr)-normalized urinary concentration of (A) IL-1β, (B) IL-8, (C) monocyte chemoattractant protein-1, and (D) IL-18. **(E, F)** Raw and Cr-normalized concentration of (E) IL-13 and (F) IL-4 as determined by multiplex immunoassay. Violin plot used to visualize the distribution and density of the data. Dotted lines show the interquartile range, and median is denoted by a horizontal dashed line. The blue circle denotes never, and the red square denotes relapse. *P*-values were generated by the Mann–Whitney *U* test.

## Determination of diagnostic cutoff concentrations for candidate urinary biomarkers

We next sought to determine the diagnostic cutoff concentrations for PGE2, IL-8, and IL-13. Cutoff concentrations of PGE2, IL-8, and IL-13 were calculated individually for the single-variable models and then sequentially for the multivariable models (Table 5). Because IL-8 was the best-performing single-variable model, we used the single-variable IL-8 posterior mean cutoff (158 pg/ml) to calculate the PGE2 cutoff. The PGE2 cutoff concentration increased from 1,808 pg/ml in the single-variable model to 1,878 pg/ml in the two-variable model (Table 5). In the three-variable model, the PGE2 and IL-8 posterior means were used to calculate the IL-13 cutoff. The IL-13 cutoff increased from 2.38 pg/ml in the single-variable

model to 2.92 pg/ml in the three-variable model (Table 5). Our results indicate that adding additional cytokines in the multivariable model does not drastically change cutoff concentrations.

## Prognostic urinary cytokines for prediction of rUTI relapse in postmenopausal women

Previously, it was demonstrated that PGE2 is not only a diagnostic UTI biomarker but also a prognostic marker of rUTI relapse (Ebrahimzadeh et al, 2021). Here, we investigated the association between urinary IL-8 and IL-13 concentration and UTI recurrence risk in the relapse cohort. We recorded time to rUTI relapse, defined as a symptomatic, culture-positive UTI, over 12 mo. To make our analysis fully comparable to the previous PGE2 study conducted in

**Table 2. Significantly differentially abundant urinary cytokines between the never and relapse groups.**

| | | IL-1β | MCP-1 | IL-8 | IL-18 | IL-13 | IL-4 |
|---|---|---|---|---|---|---|---|
| **Raw (pg/ml)** | Never | 1.5 (1.5–3.2) | 35.13 (1.9–167.4) | 7.6 (2–12.7) | 22.14 (12.4–52.5) | 1.4 (1.4–7.68) | 1 |
| | Relapse | 35.57 (7.6–152.8) | 271.6 (72.1–519) | 825.4 (130.9–1789) | 106 (69.79–319.2) | 1.4 | 1 |
| | P-value | <0.000001 | 0.000425 | <0.000001 | 0.000001 | 0.03134 | 0.227390 |
| | Fold-change | 23.4 | 7.73 | 108.6 | 4.78 | 1 | 1 |
| **Normalized (pg/µg)** | | IL-1β/Cr | MCP-1/Cr | IL-8/Cr | IL-18/Cr | IL-13/Cr | IL-4/Cr |
| | Never | 0.003 (0.001–0.01) | 0.065 (0.007–0.18) | 0.012 (0.006–0.04) | 0.0476 (0.029–0.104) | 0.0052 (0.002–0.008) | 0.00281 (0.001–0.005) |
| | Relapse | 0.058 (0.01–0.22) | 0.389 (0.21–0.49) | 0.940 (0.17–2.56) | 0.157 (0.093–0.376) | 0.0021 (0.001–0.004) | 0.00129 (0.0009–0.002) |
| | P-value | 0.000050 | 0.000078 | <0.000001 | 0.000017 | 0.002867 | 0.038516 |
| | Fold-change | 19.3 | 5.98 | 78.3 | 3.29 | 0.4 | 0.46 |

Raw and normalized median and interquartile range for each significant cytokine. P-values generated by the Mann–Whitney U test. Fold-change was calculated as a ratio of the median cytokine concentration in the relapse group and the median value in the never group.

the same cohort, urinary levels of IL-8 and IL-13 were measured in the full relapse (N = 31) and never (N = 26) groups by ELISA (Fig S5A and B) (Ebrahimzadeh et al, 2021). It was previously reported that relapse patients with above median urinary PGE2 had a higher likelihood of rUTI relapse within 1 yr (HR = 3.61, P = 0.0087) (Ebrahimzadeh et al, 2021). For IL-8, we similarly dichotomized patients about the median IL-8 concentration (489.38 pg/ml) into above (N = 15) and below median (N = 16) groups. Unlike PGE2, IL-8 had no significant predictive power for rUTI relapse (Fig 5A) (HR = 0.763, P = 0.6046). For IL-13, patients were dichotomized about the median urinary IL-13 concentration (0.41 pg/ml). Like IL-8, urinary concentration of IL-13 was not significantly associated with risk of rUTI relapse (HR = 1.028, P = 0.9546) (Fig 5B).

## Discussion

UTI is a major indication for the prescription of antibiotics, second only to respiratory tract infections (Holm et al, 2019). Overuse of antibiotics for UTI management occurs at least partly because of the lack of reliable point-of-care diagnostics (Cortes-Penfield et al, 2017). Point-of-care diagnosis of UTI currently relies on physical examination, self-reported symptoms, and urine dipstick results (Anger et al, 2019). However, differentiating UTI from other urinary tract disorders based on symptoms can be challenging. For example, although dysuria can be caused by UTI, it may also present in patients with vaginitis or chlamydial urethritis (Mambatta et al, 2015).

In addition, the urine dipstick has limited specificity with a high false-positive rate (Mambatta et al, 2015). Therefore, a reliable and accurate point-of-care diagnostic device is necessary to avoid overuse of antibiotics and improve antibiotic stewardship. However, to design such a device, accurate urinary biomarkers of UTI must be identified and validated. In this study, we aimed to identify urinary inflammatory cytokines with diagnostic power for detection

of active rUTI in a cohort of postmenopausal women. Here, we identify differentially abundant urinary cytokines in women with rUTI history and active UTI (relapse) compared with women with no history of UTI (never). We found that urinary concentrations of IL-1β, IL-8, IL-18, MCP-1 are significantly elevated in women with active rUTI compared with controls. Interestingly, a previous study of a cohort of elderly females and males reported that urinary levels of CXCL1, IL-8, and IL-6 were significantly elevated in patients with acute cystitis compared with patients with asymptomatic bacteriuria (ASB) or negative controls (Rodhe et al, 2009). It is encouraging that IL-8, one of the strongest diagnostic markers of UTI in our analysis, has been identified in multiple independent studies despite differences in study design (i.e., inclusion of males and females, different ages, and comparison to ASB) (Ko et al, 1993; Rao et al, 2001; Oregioni et al, 2005). In addition, Ko et al demonstrated that urinary levels of IL-8 were higher than serum levels suggesting local production and supporting the use of urinary cytokines as diagnostic markers (Ko et al, 1993).

We also found that IL-4 and IL-13, which are two anti-inflammatory cytokines with shared immunoregulatory effects, were reduced in the relapse group. IL-13, for example, can inhibit the production of pro-inflammatory cytokines, including IL-1α, IL-1β, IL-6, IL-8, G-CSF, and IFN-α (de Vries, 1998). Using logistic regression analysis, we determined that a three-variable model consisting of IL-8, PGE2, and IL-13 reliably discriminated patients based on current UTI status. Lastly, we found that unlike urinary PGE, elevated urinary IL-8 and IL-13 concentrations were not associated with an increased risk of rUTI relapse within 12 mo.

This research provides rationale for future studies to further validate urinary IL-8, PGE2, and IL-13 as immune diagnostic biomarkers of UTI and PGE2 as a rUTI prognostic biomarker in larger and more diverse cohorts including those with ASB and acute cystitis. Also, the differences in UTI diagnostic cytokines identified between studies in different patient groups highlights the need for the development of more specialized diagnostic panels for

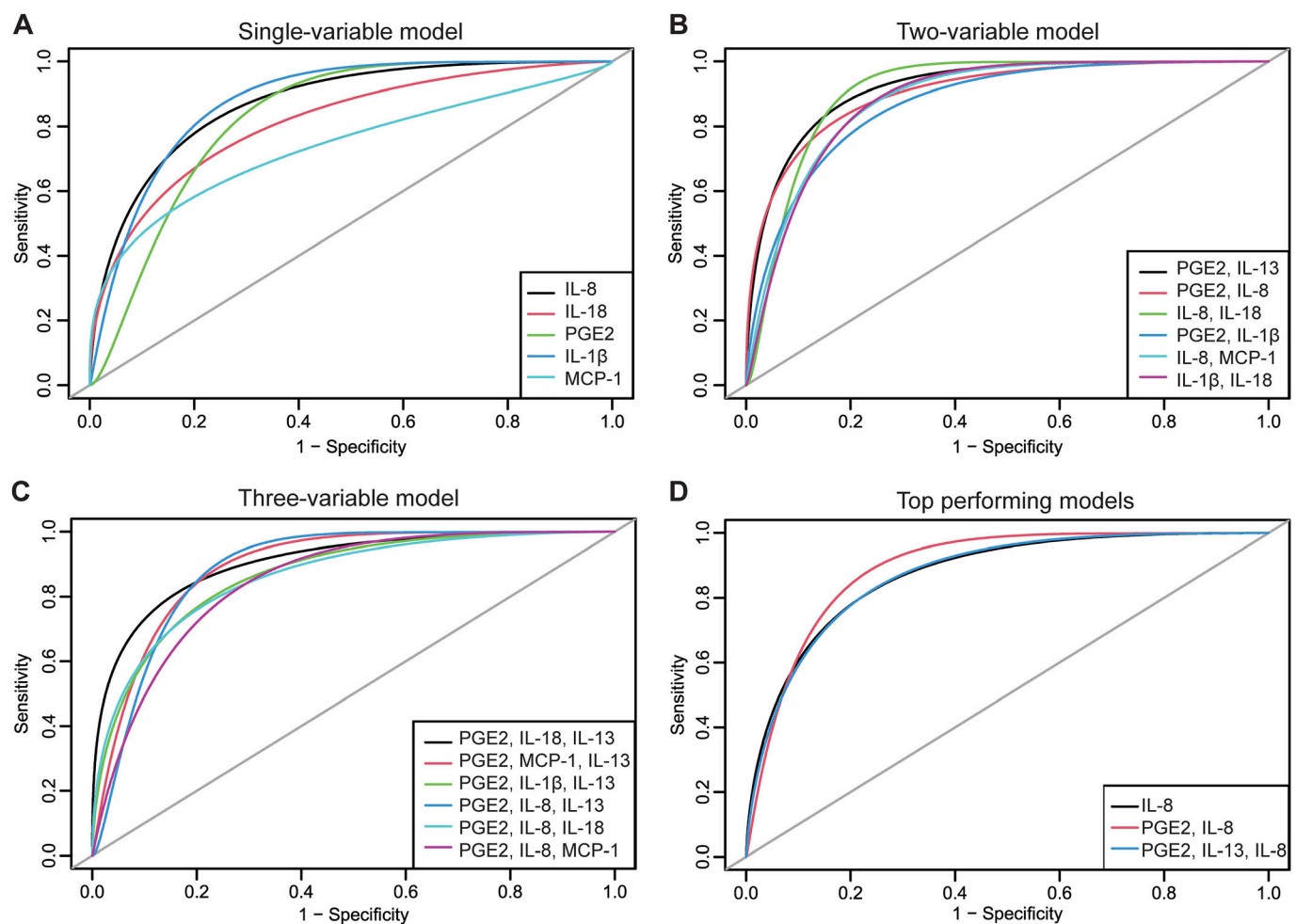

**Figure 4. Top-performing urinary cytokine–based urinary tract infection diagnostic models.**
**(A, B, C)** Receiver operating characteristic curves of (A) single-variable, (B) two-variable, and (C) three-variable logistic regression models demonstrate predictive power to discriminate patients based on their urinary tract infection status. **(D)** Comparison of the top-performing model and single-, two-, and three-variable diagnostic models.

**Table 3. Single-variable logistic regression analysis of candidate cytokines.**

| Variable | AUC | Misclassification rate | Sensitivity | Specificity | Pseudo-$R^2$ | *P*-value |
|---|---|---|---|---|---|---|
| IL-8 | 0.885 | 0.152 | 0.739 | 0.957 | 0.476 | 0.000 |
| IL-18 | 0.836 | 0.239 | 0.652 | 0.870 | 0.267 | 0.000 |
| PGE$_2$ | 0.832 | 0.217 | 0.739 | 0.826 | 0.289 | 0.000 |
| IL-1β | 0.815 | 0.174 | 0.696 | 0.957 | 0.342 | 0.000 |
| MCP-1 | 0.732 | 0.283 | 0.652 | 0.783 | 0.219 | 0.000 |
| IL-13 | 0.293 | 0.391 | 0.913 | 0.304 | 0.092 | 0.015 |
| IL-4 | 0.166 | 0.435 | 0.957 | 0.174 | 0.035 | 0.135 |

Metrics including area under the curve, misclassification rate, sensitivity, and specificity were attained through leave-one-out cross-validation. McFadden's pseudo-$R^2$ and its *P*-value were used to assess the fitness of each model.

different demographic or clinical groups (Ambite et al, 2016; Sundac et al, 2016; Armbruster et al, 2018). Identification of highly accurate urinary cytokine panels for UTI diagnosis will be critical for innovation in the field of UTI diagnostics because these cytokine-based urinary UTI biomarkers may be useful in the development of new point-of-care (POC) devices for UTI diagnosis. For example, Jagannath et al have reported a wearable SWEATSENSER which can detect infection through the cytokines found in eccrine sweat

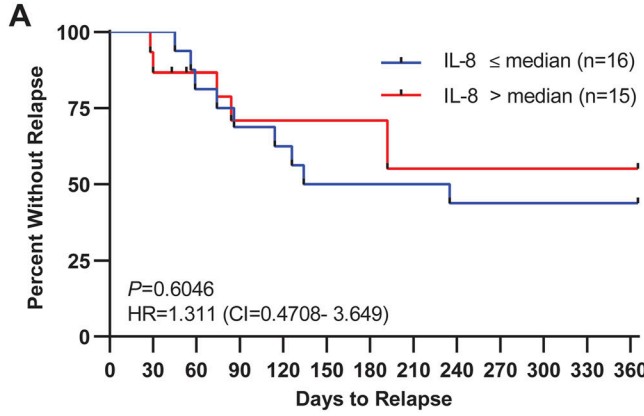

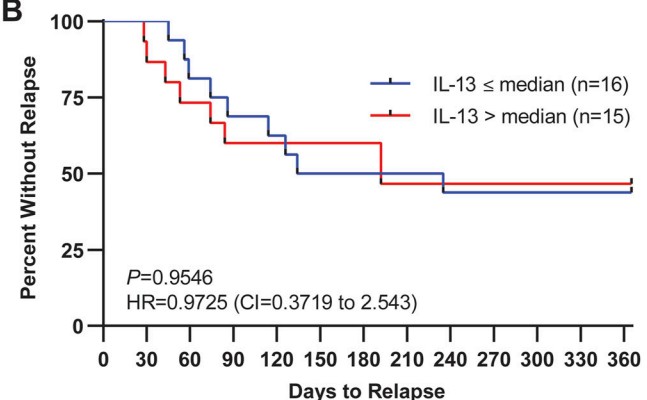

**Figure 5. IL-8 and IL-13 do not predict recurrent urinary tract infection relapse in postmenopausal women.**
**(A, B)** Kaplan–Meier analysis of time to relapse of relapsed patients dichotomized about median (A) IL-8 or (B) IL-13 concentration. The red line depicts above median and the blue line below median. Y-axis depicts the percentage of patients with no urinary tract infection episode. X-axis shows the follow-up period (days). Data were analyzed by the log-rank (Mantel–Cox) test. HR, hazard ratio (below median/above median) and CI = 95% confidence interval.

(Jagannath et al, 2021). Validation of these or other urinary cytokines may facilitate the development of similar sensors into POC UTI diagnostic tools (Ganguly et al, 2023).

One limitation of a cytokine-based UTI diagnosis is that it does not include antimicrobial susceptibility testing and thereby may not reduce the use of broad-spectrum antibiotics. However, a rapid, cytokine-based POC diagnostic device may serve as effective triage to identify individuals with potential symptomatic UTI versus those with ASB or a noninfective condition, especially in populations who cannot effectively communicate symptoms, ahead of non-POC tests that include antimicrobial susceptibility testing. To further distinguish ASB, it may also be important to

include bacterial cell surface markers as a parameter to be measured along with pro- and anti-inflammatory cytokines (Stapleton et al, 2015). Furthermore, a POC diagnostic device that can predict risk of rUTI relapse may allow for more personalized treatment plans aimed at reducing that risk. Once independently validated, these biomarkers may be implemented in the development of new POC diagnostic devices for rapid and accurate detection of UTI and rUTI prognosis.

# Materials and Methods

Patient samples were collected between May 2018 and June 2020 after informed patient consent and Institutional Review Board approval (STU 082010-016, STU 032016-006, MR 17-120).

### Cohort design, sample collection, and classification

Samples were selected from 245 archived midstream clean catch urine samples that were previously obtained from consenting patients at the University of Texas Southwestern Medical Center Urology Clinic. Urine was immediately chilled, aseptically processed, and stored in liquid nitrogen within 2 h. 57 patients passed the exclusion criteria of pre- or peri-menopausal, sporadic UTI defined as a single UTI in the previous year (Bradley et al, 2020), PVR > 150 ml, >stage two bladder prolapse, immune suppression, neurogenic bladder, history of catheterization, and surgery less than a month prior sample collection. Patients were classified as either no UTI history (no clinical history of symptomatic UTI, N = 26) or relapse rUTI (history of rUTI with a current, symptomatic UTI, N = 31). History of UTI was documented through patient survey and chart review, and rUTI was defined as two UTIs in the previous 6 mo or three in the previous year. UTI symptoms were recorded by the previously validated UTISA lower urinary tract symptoms questionnaire (Clayson et al, 2005).

### Urinalysis

The automated point-of-care urine analyzer CLINITEK Status+ (Siemens) was used to perform dipstick urinalysis (UA) on relapse rUTI patients. The reagent strip tests for protein, blood, LE, Ni, glucose, ketone, pH, specific gravity, bilirubin, and urobilinogen. Patients were considered UA-positive if LE and/or Ni were greater than cutoff value of +1. Blood and protein were positive if >trace.

**Table 4. Summary of top-performing urinary cytokine–based urinary tract infection diagnostic models.**

| Variable | AUC | Misclassification rate | Sensitivity | Specificity | F-score | Pseudo-$R^2$ |
|---|---|---|---|---|---|---|
| IL-8 | 0.885 | 0.152 | 0.739 | 0.957 | 0.884 | 0.476 |
| PGE2, IL-8 | 0.904 | 0.152 | 0.826 | 0.870 | 0.880 | 0.601 |
| PGE2, IL-13, IL-8 | 0.905 | 0.109 | 0.913 | 0.870 | 0.913 | 0.797 |

**Table 5.** Cytokine cutoff concentrations calculated from Bayesian logistic regression models.

| Cytokines | Intercept | Coefficient | Cutoff (pg/ml) |
|---|---|---|---|
| IL-8 | −1.6128 | 0.0113 | 158.21 |
| PGE2 | −2.8144 | 0.0016 | 1808.05 |
| IL-13 | 0.7809 | −0.3214 | 2.38 |
| Two-variable model | −4.127 | | |
| IL-8 | | 0.009 | |
| PGE2 | | 0.001 | 1878.03 |
| Three-variable model | −6.442 | | |
| PGE2 | | 0.004 | |
| IL-8 | | 0.10 | |
| IL-13 | | −1.20 | 2.92 |

## Urine culture (UC)

Clinical UC was performed on all relapse rUTI patients by the UTSW Clinical Microbiology Laboratory. Patients with rUTI history and UTI symptoms but clinical UC reported bacteriuria below the UC cutoff of ≤$10^5$ CFU/ml were cultured by more sensitive methods. 100 $\mu$l of urine was plated on BBL CHROMagar Orientation and incubated at 37°C for 24–72 h. >$10^3$ CFU/ml was considered UC-positive. The bacteriology chart provides information on a single bacterial species isolated from the first clinical culture of relapse patients. All relapse samples were subjected to UC and no extended UC techniques.

## Urinary cytokine screening

A priori power analysis was performed to compute the minimum sample size (N = 46) required to detect an effect size of 0.5 with a power of 0.9 (Fig S6). Urinary levels of pro-inflammatory cytokines IL-1$\beta$ (LoD 1.5 pg/ml), IL-5 (LoD 1.2 pg/ml), IL-6 (LoD 1 pg/ml), IL-8 (LoD 2 pg/ml), IL-9 (LoD 1.7 pg/ml), MCP-1 (LoD 1.1 pg/ml), IFN-$\gamma$ (LoD 1.1 pg/ml), IL-12p70 (LoD 2 pg/ml), IL-17A (LoD 1.9 pg/ml), IL-17F (LoD 0.8 pg/ml), IL-18 (LoD 1.3 pg/ml), IL-22 (LoD 1.5 pg/ml), IL-23 (LoD 1.8 pg/ml), and IL-33 (LoD 4.4 pg/ml) and anti-inflammatory cytokines IL-4 (LoD 1.5 pg/ml), IL-10 (LoD 0.7 pg/ml), and IL-13 (LoD 1.4 pg/ml) were measured using the LEGENDplex HU Th Cytokine Panel (12-plex) and the LEGENDplex Human Inflammation Panel 1 (8-plex) bead-based immunoassays followed by flow cytometry. Urine samples were diluted 1:2 in assay buffer before analysis. Data acquisition was completed on a BD Fortessa flow cytometer. LEGENDplex Data Analysis software calculated the concentration of each analyte.

## Creatinine measurement

Urinary creatinine measurement was described previously (Ebrahimzadeh et al, 2021). Briefly, urinary creatinine was measured with the Creatinine Urinary Detection Kit (Thermo Fisher Scientific) at optical density 490 nm using the Synergy H1 plate reader (BioTek).

## IL-8 and IL-13 ELISA

Quantification of urinary IL-8 (LoD 1 pg/ml) and IL-13 (LoD 0.41 pg/ml) was performed using the Human IL-8/CXCL8 ELISA Kit and Human IL-13 ELISA Kit (Sigma-Aldrich). Optical density was measured at 450 nm with a Synergy H1 plate reader (BioTek).

## Statistical analysis

Statistical analyses were performed with GraphPad Prism 8.1.0 and RStudio version 4.0.2. Two-sided parametric ($t$ and $\chi^2$) and non-parametric (Mann–Whitney) tests were used at $\alpha$ = 0.05 to test for group differences. The Z-score was calculated through Prism using the formula $z = \frac{x-\mu}{\sigma}$ where x is the raw score, $\mu$ is the population mean, and $\sigma$ is the SD. Logistic regression was used to determine the relationship between rUTI and cytokine concentration with a standard cutoff probability of 0.5. McFadden's pseudo-$R^2$ and its $P$-value were used to assess model fit. Leave-one-out cross-validation was used to evaluate the predictive power of each logistic model via AUC, sensitivity, specificity, misclassification rate, and F1-score. Receiver operating characteristic curves were generated by plotting the false-positive rate (1-specificity) against the true-positive rate (sensitivity).

## Cytokine cutoff calculation using Bayesian methodology

We used a Bayesian logistic regression model to estimate cytokine cutoffs and credible intervals. The logit function was defined as $\log\frac{\pi}{1-\pi} = \boldsymbol{X\beta}$ where $\boldsymbol{\beta} = (\beta_0, \beta_1, …, \beta_p)^\top$ was the vector containing $p$ + 1 model coefficients, $\boldsymbol{X} = (\mathbf{1}, \boldsymbol{x}_1, …, \boldsymbol{x}_p)$ was the $n \times (p + 1)$ design matrix with $x_j = (x_{1j}, …, x_{ij}, …, x_{nj})^\top$ as the cytokine measurements for sample $i$ = 1, …, $n$ and cytokine $j$ = 1, …, $p$. Likelihood was modeled by a Bernoulli distribution with the probability parameter $\pi = \frac{e^{\boldsymbol{X\beta}}}{1 + e^{\boldsymbol{X\beta}}}$, and we assigned a $t$-distribution as the prior distribution for each model coefficient. The posterior distribution was a product of Bernoulli and t-distributions. Each model was fit using a random walk Metropolis–Hastings MCMC algorithm. The values of the model coefficients were estimated by their respective posterior means, and acceptance rates for all models were between the desired range of 0.2 and 0.5.

To determine the cutoff values, we set the cutoff probability of the logit function to $\pi$ = 1/2 and solved the resulting linear equation $\boldsymbol{X\beta}$ = 0. For single-variable models, cutoffs were defined by $x_1 = -\beta_0/\beta_1$. For two- and three-variable models, cutoffs were determined sequentially, taking the cutoff for the cytokine having the highest AUC in the single-variable model as $x_1$. The cutoff of the second variable was calculated using $x_2 = (-\beta_0 - \beta_1 x_1)/\beta_2$. Posterior mean and 95% credible interval were calculated for each value.

## Time-to-relapse analysis

UTI episodes were recorded in a 12-mo follow-up study in which patients in the relapse cohort were followed through chart review and phone calls. Kaplan–Meier analysis was performed to assess the difference in the risk of rUTI relapse between the two groups. The Mantel–Cox log-rank test was used to calculate the hazard ratio (HR).

## Data Availability

The data supporting the conclusions of this study that are not already provided in the main text and supplemental information and are subject to patient confidentiality protections will be shared upon reasonable request.

## Supplementary Information

## Acknowledgements

The authors would like to sincerely thank the women who participated in this study. This work was supported by grants from the Welch Foundation (AT-2030-20200401) and the National Institutes of Health (1R01DK131267-01) to NJ De Nisco and by the Felicia and John Cain Distinguished Chair in Women's Health to PE Zimmern.

### Author Contributions

T Ebrahimzadeh: conceptualization, formal analysis, investigation, visualization, methodology, and writing—original draft.
U Basu: formal analysis, investigation, visualization, and writing—original draft, review, and editing.
KC Lutz: formal analysis, visualization, methodology, and writing—original draft, review, and editing.
J Gadhvi: formal analysis, investigation, and methodology.
JV Komarovsky: investigation.
Q Li: conceptualization, supervision, and validation.
PE Zimmern: conceptualization, supervision, funding acquisition, and writing—review and editing.
NJ De Nisco: conceptualization, supervision, funding acquisition, validation, methodology, project administration, and writing—original draft, review, and editing.

### Conflict of Interest Statement

The authors declare that they have no conflict of interest.

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
