## [Reviewer comments · Life Science Alliance]

Life Science Alliance

Inflammatory markers for improved recurrent UTI diagnosis in postmenopausal women

Tahmineh Ebrahimzadeh, Ujjaini Basu, Kevin C . Lutz, Jashkaran Gadhvi, Jessica V. Komarovsky, Qiwei Li, Philippe E. Zimmern, and Nicole J. De Nisco

DOI: <https://doi.org/10.26508/lsa.202302323>

Corresponding author(s): *Nicole J. De Nisco, The University of Texas at Dallas*

Review Timeline:

Submission Date:	2023-08-15
Editorial Decision:	2023-09-26
Revision Received:	2024-01-03
Editorial Decision:	2024-01-19
Revision Received:	2024-01-30
Accepted:	2024-01-31

Transaction Report:

September 26, 2023

Re: Life Science Alliance manuscript #LSA-2023-02323

Dr. Nicole J De Nisco
The University of Texas at Dallas
Biological Sciences
800 W. Campbell Road
BSB12.515
Richardson, Texas 75080

Dear Dr. De Nisco,

Thank you for submitting your manuscript entitled "Inflammatory markers for improved recurrent urinary tract infection diagnosis in women" to Life Science Alliance. The manuscript was assessed by expert reviewers, whose comments are appended to this letter. We invite you to submit a revised manuscript addressing the Reviewer comments.

Thank you for this interesting contribution to Life Science Alliance. We are looking forward to receiving your revised manuscript.

Sincerely,

B. MANUSCRIPT ORGANIZATION AND FORMATTING:

Reviewer #1 (Comments to the Authors (Required)):

The ability to quickly and accurately detect UTI is imperative for rapid diagnostics at point-of-care, followed by the administration of (ideally, targeted) antibiotics. Almost all new diagnostic development seems to be focused on the pathogen - which is difficult considering the prevalence of polymicrobial infections and the presence of the urinary microbiota, confounded by the high sensitivity of genomics-based tests. Therefore host-based tests are a promising area, especially those that would detect infection despite the vast diversity of species and strains that can cause a UTI. The world still awaits a better replacement for the much-maligned dipstick, which is cheap and easy but almost entirely useless. This is a focused, well-written and well-designed study which builds on the same authors' work with PGE2 with some promising findings. The authors have used a cohort of non-UTI vs recurrent UTI patients to show that a particular suite of cytokines, namely PGE2, IL-8, and IL-13, can distinguish between patients and controls with good sensitivity and specificity.

Data are largely supportive for all main points that the authors make (with the possible exception of IL4, see below specific comments).

Below are some comments that might lead to improvement of this interesting paper.

1. There seems to be some mixed messaging here: are the authors suggesting their cytokine fingerprint should be for predicting (A) likelihood of recurrence (as per 'Prognostic' section starting line 197), or for (B) diagnosing a current infection? Or both? Some clarity in the Discussion would be helpful. Has a case truly been made for why this is superior to the diagnosis already made to put them into the rUTI group in the first place?
2. If it's (A), if they have a history of UTI, they are already known to be rUTI sufferers. How would the new test be used clinically? If they are at higher risk what could a clinician do for them practically? Many such women are already on prophylactic antibiotics and now they are starting to try vaccines. Will a future test based on this really be cost-effective if there's not much to be done with the knowledge?
3. If it's primarily for (B), have the authors really demonstrated it's superior to the way they chose their cohort in the first place? A bit of a circular logic here: they chose their rUTI cohort based on the traditional diagnostic methods and then validated the new cytokine analysis against that. By definition they excluded the very patients who might have benefitted from the test - e.g culture-negative non-UTI individuals with copycat symptoms, or culture-"negative" (below the traditional threshold or mixed growth) UTI individuals - we don't know if this new suite of analytes will ignore the former or pick out the latter. An arm of culture-negative suspected UTI patients would have been truly interesting - though this referee is not unaware of the difficulties in defining such patients, and probably this is something for a follow-up study.
4. The authors point to Oregioni in the Intro. Do they know if particular cytokines are associated with specific pathogen species in the patients in their own study or whether the suite of analytes is a "pan" detector of UTI regardless of species?
5. I am a bit wary of the IL-4 conclusion. Just from eyeballing the graph it looks as if one outlier may have dragged the P value down to that rather underwhelming 0.03... Given that it isn't important for the ultimate fingerprint suite of analytes, is this really a hill you want to die on?
6. Sorry to be nit-picky, but is this really only a 'retrospective' study considering the 12-month follow-up stage? It does have some elements of a prospective study...perhaps alter the wording somewhat?
7. Line 30 - the 150 million per year figure is a bit outdated now. Latest is 400 million (Yang et al, 10.3389/fpubh.2022.888205).
8. Line 79 - should it actually be 'functional orthologues'?
9. Line 106. Could the fulguration technique lead to some long-term irritation/inflammation (with specific cytokines associated) as presumably it damages the urothelium? If this is a potential confounder, is the sample size large enough to do subgroup analysis?
10. Line 115 and elsewhere, should be x % "were" positive, not "was" positive
11. Line 118+, did the patients really have only one pathogen? Is this % of patients with this bug only, or this plus others? Very unusual for people in this age group to only grow one especially as they dropped down to 10^3 and grew for longer on the more amenable media. If not, how do we know if all of these are infections/pathogens and not commensals? Would be nice to have a table of all species that grew in each patient and their relative frequencies in supplemental for those interested in such things. As a related point, it seems a missed opportunity not to do bacteriology on the controls (as per Fig 2C), as an understanding of any shifts in urobiome, correlated with immune response, would have been interesting and would also make the first point ('what

actually is the pathogen') somewhat clearer.

12. Line 197, typo in the word post(m)enopausal

13. Line 202. "To increase the rigor of this analysis and to make it fully comparable to the previous PGE2 study... urinary levels ... were measured in the full Relapse (N=31) and Never (N=26) groups by enzyme-linked immunoassay (ELISA)"

This is confusing for people who don't know detailed pros and cons of ELISA vs multiplexed immunoassays - at first glance, one wouldn't think that the latter are less "rigorous" than the former. Might need a line explaining this choice better (as people might then think that Figure 3 results aren't actually rigorous). Also, add mention of the multiplex method to Fig 3 legend title to parallel that of S3 which mentions ELISA in the title.

14. Line 236 - It's "interesting", but also authors could contextualize their results more by perhaps speculating on differences in study design that might have led to the current study not picking up IL6 as Rodhe et al did (perhaps a general comment about how different studies of this type have implicated different sets of cytokines)? Why is IL8 not found in so many other studies?

15. Authors should mention the limitations inherent in utility of inflammatory markers for POC diagnostics when the goal in many countries is to avoid the use of empirical broad-spectrum antibiotics - which a test based on the current suite of cytokines would not help. Perhaps this needs to be used in conjunction with a pathogen-focused test (ideally one with antimicrobial sensitivity testing baked in) for a truly holistic test.

Reviewer #2 (Comments to the Authors (Required)):

This manuscript by Ebrahimzadeh et al. investigates an important clinical issue, recurrent urinary tract infection. Urinary cytokines were measured in postmenopausal women with either rUTI or no recent UTI history with the goal of identifying a cytokine profile that could predict rUTI status and be developed as a point-of-care assessment. The study is a follow-up to a previous publication from the same group (Ebrahimzadeh et al. 2021), where PGE2 was found to be a marker that both signaled current rUTI and was predictive of future episodes. The data provided in the current manuscript provide a more refined diagnostic for rUTI over the previous manuscript. Although the longitudinal follow-up of the patient cohort is a strength, it was disappointing that the identified markers were not predictive of outcomes. The manuscript is well written and the conclusions were straightforward.

I have one major concern: prior work on the identified cytokines in human populations needs to be more prominently stated to properly establish the context of this study. Specifically, IL-8 was found to be elevated in patient UTI urine samples at least as early as 1993 (PMID: 8454332, not cited), and has been previously proposed as a UTI diagnostic marker, although this cytokine may be elevated for other reasons (PMID: 11517116, also not cited). Thus, the advances in the current study may be more specifically applied to rUTI in postmenopausal women (an understudied cohort), and the major strength is the combinatorial panel of cytokines that was identified as a useful diagnostic. The inclusion of a marker that is elevated in the non-UTI "Never" group, IL-13, is especially interesting.

Minor comments follow.

1. Title: specify "postmenopausal" women

2. Line 98: would it be more accurate to state the "never" group has no recent UTI history? Based on the Methods (lines 266-7), it appears that patients with recent UTI (1x in previous year) were classified as "sporadic" and excluded from the study. It would follow that the "never" group would likely include women with prior UTI history of over 1 year ago.

3. Line 121/Fig 2C: it would be helpful to include absolute numbers in Fig. 2C along with the percentages. For example, it would be clear that 3.2% is n=1.

4. Line 143: specify the limit of detection, either here or in the Methods.

5. Line 197: typo "postmenopausal"

6. Lines 233-6: please elaborate on why this finding is interesting. Only one of the cytokines from this study mentioned in the previous sentence was elevated in Rodhe et al. Were all the same cytokines tested in both studies? Was something unexpected? Should we conclude or expect that some of the mentioned cytokines will be more indicative of uncomplicated cystitis vs. rUTI?

7. Line 253: please clarify the statement about LPS. First, LPS would be generically present for Gram-negative bacteria and not at all for Gram-positive bacteria or non-bacterial UTI pathogens, regardless of UTI vs ASB. Second, I could not find mention of "lipopolysaccharide," "LPS," or "endotoxin" in the Stapleton et al. reference provided and it may have been mis-cited. I am aware of studies stating some LPS are more inflammatory than others (e.g., David Klumpp's work), although how this would be incorporated into a diagnostic kit is not clear from this discussion point.

8. Lines 280-4 Methods and Fig. 2C: please clarify if only monospecies infections were used for classifying bacteriology.

9. Please check that the supplementary tables display correctly in the final version. On my review copy, many items were obscured where lines wrapped and were cut off or superimposed.

Reviewer #3 (Comments to the Authors (Required)):

Ebrahimzadeh et al submit new results of pro-inflammatory cytokine profiling of urine specimens from postmenopausal women with and without history of recurrent UTI. Recurrent UTI is a major medical burden for this population, and current clinical

diagnostic methods (urine culture) often take >24 hours to obtain results. In this work, the authors used a retrospective clinical cohort to identify subjects with documented rUTI (and UTI at the time of sampling) and subjects with no history of UTI and UTI negative at the time of sampling. Using multiplex cytokine arrays and follow-up targeted ELISA, the authors found that a multivariate model of IL-8, PGE2 (measured in a prior study), and IL-13 displayed the highest specificity and sensitivity for rUTI prediction. This is an important finding for designing improved diagnostics for rUTI in postmenopausal patients. Overall, the study is well-designed, data is analyzed and interpreted appropriately, and the manuscript is well-written. One concern is the lack of clarity on whether all urine samples were subjected to urine culture or extended urine culture techniques. Additional specific comments are focused on areas that may benefit from additional discussion or clarification of the presented work.

Specific comments:

1. Lines 98-99: Unclear why an additional arm - spontaneous UTI (not rUTI) was not included in analyses? Although the ability to identify rUTI biomarkers in postmenopausal women is of clinical value, the ability to also detect UTI in this population would also be valuable.
2. Lines 101-103: Given the differences in age and BMI between cohort arms, were age and BMI evaluated as variables in cytokine analyses?
3. Lines 98-99, lines 265-269, and lines 280-281: It is unclear whether all samples were cultured as a part of urine collection. Line 280 states that UC was performed on Relapse rUTI patients. Was UC/extended UC performed on never UTI patients to account for ASB? Urine culture (and perhaps dipstick) should be performed on samples from patients in the Never group to confirm absence of urinary bacteria. Patient reported symptoms would not delineate between patients with negative bacterial culture and patients with asymptomatic bacteruria. Further, comparing the specificity of urine dipstick to urinary cytokines in this cohort could further enhance the predictive power of this model.
4. Figure S2 - consider performing hierarchical clustering to identify co-occurring cytokines. Also, please provide details on how z scores calculated. Do cytokine patterns correlate with pathogens cultured from urine? Do patients with UPEC have altered cytokine profiles compared to non-UPEC cases?
5. Table 2/Figure 3/Figure S3 - Indicate limits of detection for each cytokine. Samples following below the limit of detection should be displayed below the limit of detection (indicate n of these samples). Samples falling below the limit of detection should not be carried into normalized values since nothing was detected.
6. Line 210: Has a multivariate model been applied here to determine if the pairing PGE2 and IL-8 increases the predictive power for recurrence?
7. Lines 233-236: a limitation of this study is that only rUTI and never UTI samples were compared. Extending these analyses to acute cystitis and ASB samples would be important future work - especially given the elevated rate of ASB in elderly, postmenopausal women.

Response to reviewers LSA-2023-02323R

We thank the reviewers for their thorough review of our manuscript and their thoughtful feedback. We have addressed all reviewer comments and made the requested changes. Please find our point by point responses to the reviewer comments in blue below:

Reviewer #1 (Comments to the Authors (Required)):

The ability to quickly and accurately detect UTI is imperative for rapid diagnostics at point-of-care, followed by the administration of (ideally, targeted) antibiotics. Almost all new diagnostic development seems to be focused on the pathogen - which is difficult considering the prevalence of polymicrobial infections and the presence of the urinary microbiota, confounded by the high sensitivity of genomics-based tests. Therefore host-based tests are a promising area, especially those that would detect infection despite the vast diversity of species and strains that can cause a UTI. The world still awaits a better replacement for the much-maligned dipstick, which is cheap and easy but almost entirely useless. This is a focused, well-written and well-designed study which builds on the same authors' work with PGE2 with some promising findings. The authors have used a cohort of non-UTI vs recurrent UTI patients to show that a particular suite of cytokines, namely PGE2, IL-8, and IL-13, can distinguish between patients and controls with good sensitivity and specificity.

Data are largely supportive for all main points that the authors make (with the possible exception of IL4, see below specific comments).

Below are some comments that might lead to improvement of this interesting paper.

1. There seems to be some mixed messaging here: are the authors suggesting their cytokine fingerprint should be for predicting (A) likelihood of recurrence (as per 'Prognostic' section starting line 197), or for (B) diagnosing a current infection? Or both? Some clarity in the Discussion would be helpful. Has a case truly been made for why this is superior to the diagnosis already made to put them into the rUTI group in the first place?

We apologize for the confusion. Here we find that PGE2, IL-8, and IL-13 are accurate for diagnosis of UTI in the cohort of women analyzed in this manuscript. We had previously found that women with elevated PGE2 had a >3x greater risk of UTI relapse. Interestingly, we tested if IL-8 and IL-13 would be similarly predictive in the current manuscript and still PGE2 remained the only prognostic biomarker able to predict rUTI relapse. We have added text starting on line 267 of the marked up copy of the manuscript to clarify this point.

2. If it's (A), if they have a history of UTI, they are already known to be rUTI sufferers. How would the new test be used clinically? If they are at higher risk what could a clinician do for them practically? Many such women are already on prophylactic antibiotics and now they are starting to try vaccines. Will a future test based on this really be cost-effective if there's not much to be done with the knowledge?

We think it would be clinically useful to be able to stratify patients based on their risk for recurrent UTI. This would allow for the development of more personalized treatment plans

for those patients with higher risk scores aimed at reducing this risk, for example, adjunct therapy with NSAIDs or a longer course of antibiotic therapy. Again, as mentioned in the discussion, these findings are preliminary and need to be validated in larger cohorts, but if elevated PGE2 is mechanistically involved in increased rUTI susceptibility (as has been shown in mouse models) then it may be a viable target for new rUTI treatment paradigms. We have added text to the second to last paragraph in the discussion clarifying this point.

3. If it's primarily for (B), have the authors really demonstrated it's superior to the way they chose their cohort in the first place? A bit of a circular logic here: they chose their rUTI cohort based on the traditional diagnostic methods and then validated the new cytokine analysis against that. By definition they excluded the very patients who might have benefitted from the test - e.g culture-negative non-UTI individuals with copycat symptoms, or culture-"negative" (below the traditional threshold or mixed growth) UTI individuals - we don't know if this new suite of analytes will ignore the former or pick out the latter. An arm of culture-negative suspected UTI patients would have been truly interesting - though this referee is not unaware of the difficulties in defining such patients, and probably this is something for a follow-up study.

Thank you for this comment. We are using urine culture and symptoms as the gold standard and not trying to show that these immune biomarkers are more accurate than the gold standard in this manuscript. Urine culture has its own problems (likely a higher false negative rate due to the limited culture conditions) and cannot be developed into a point-of-care diagnostic. We are comparing the predictive value of these urinary cytokines for UTI diagnosis instead to reported values for the existing urine dipstick with the goal of finding better immune biomarkers for integration into a point-of-care dipstick device. We agree that it would be interesting to recruit a group of culture-negative susceptible UTI patients for a follow-up study.

4. The authors point to Oregioni in the Intro. Do they know if particular cytokines are associated with specific pathogen species in the patients in their own study or whether the suite of analytes is a "pan" detector of UTI regardless of species?

As per the authors suggestion we tested if any cytokines were associated with a specific bacterial species and we did not find any association. We report these data in Figure S4 and have referenced this finding in the results section.

5. I am a bit wary of the IL-4 conclusion. Just from eyeballing the graph it looks as if one outlier may have dragged the P value down to that rather underwhelming 0.03... Given that it isn't important for the ultimate fingerprint suite of analytes, is this really a hill you want to die on?

We agree about the data for IL-4. We have accordingly been very measured in our description of these findings. Starting in line 146 of the results section we say that normalized IL-4 is only slightly higher and that IL-4 was under the limit of detection for most patients. We have tried to be very transparent in the reporting of our data to avoid misinterpretation.

6. Sorry to be nit-picky, but is this really only a 'retrospective' study considering the 12-month follow-up stage? It does have some elements of a prospective study...perhaps alter the wording somewhat?

We have removed the word "retrospective" from line 98.

7. Line 30 - the 150 million per year figure is a bit outdated now. Latest is 400 million (Yang et al, 10.3389/fpubh.2022.888205).

Thank you for this updated number! We have revised the sentence and included the new reference.

8. Line 79 - should it actually be 'functional orthologues'?

We have revised this as requested.

9. Line 106. Could the fulguration technique lead to some long-term irritation/inflammation (with specific cytokines associated) as presumably it damages the urothelium? If this is a potential confounder, is the sample size large enough to do subgroup analysis?

We attempted the subgroup analysis as suggested and the results are in Figure S5 A,B. There was no significant difference between patients with no prior fulguration and those with a prior fulguration in the Relapse group.

10. Line 115 and elsewhere, should be x % "were" positive, not "was" positive

Changed.

11. Line 118+, did the patients really have only one pathogen? Is this % of patients with this bug only, or this plus others? Very unusual for people in this age group to only grow one especially as they dropped down to 10^3 and grew for longer on the more amenable media. If not, how do we know if all of these are infections/pathogens and not commensals? Would be nice to have a table of all species that grew in each patient and their relative frequencies in supplemental for those interested in such things. As a related point, it seems a missed opportunity not to do bacteriology on the controls (as per Fig 2C), as an understanding of any shifts in urobiome, correlated with immune response, would have been interesting and would also make the first point ('what actually is the pathogen') somewhat clearer.

We have added a new supplemental table (Table S1) that contains the clinical culture results for each of the relapse patients. The pie chart displays the major (most abundant pathogen), but as you can see from the new Table S1, instances of secondary species were common with *E. coli* and *E. faecalis* commonly co-occurring. There are of course limitations to clinical urine culture methods as they rely on a small volume of urine and very specific culture conditions, so it is possible that there are other microbiota underlying these pathogens as well that were not detected by clinical urine culture. We have revised the figure legend to

specify that the species labelled in the pie chart were the major species detected and that the full bacteriology can be found in Table S1.

12. Line 197, typo in the word post(m)enopausal

Changed.

13. Line 202. "To increase the rigor of this analysis and to make it fully comparable to the previous PGE2 study... urinary levels ... were measured in the full Relapse (N=31) and Never (N=26) groups by enzyme-linked immunoassay (ELISA)"

This is confusing for people who don't know detailed pros and cons of ELISA vs multiplexed immunoassays - at first glance, one wouldn't think that the latter are less "rigorous" than the former. Might need a line explaining this choice better (as people might then think that Figure 3 results aren't actually rigorous). Also, add mention of the multiplex method to Fig 3 legend title to parallel that of S3 which mentions ELISA in the title.

Thanks for pointing this out. We removed the phrase "to increase rigor" and just kept to make it fully comparable to avoid confusion. Now it should communicate the fact that we wanted to compare data generated by the same methods. We also added mention of the multiplex method in the Figure 3 legend text since the title was already long.

14. Line 236 - It's "interesting", but also authors could contextualize their results more by perhaps speculating on differences in study design that might have led to the current study not picking up IL6 as Rodhe et al did (perhaps a general comment about how different studies of this type have implicated different sets of cytokines)? Why is IL8 not found in so many other studies?

We think it is actually encouraging that IL-8 was found both in our study and the Roche study despite the difference in study design. We have added a sentence to communicate this in the discussion. We do recognize that different studies of this type seem to show different results and believe this can be due to differences in study design as well as in patient demographics and clinical characteristics. Specific demographic or clinical groups may have their own unique set of best performing diagnostic urinary cytokines. These differences highlight the need for larger and more diverse studies in this area. We have revised the beginning of the fourth paragraph of the discussion to better communicate this.

15. Authors should mention the limitations inherent in utility of inflammatory markers for POC diagnostics when the goal in many countries is to avoid the use of empirical broad-spectrum antibiotics - which a test based on the current suite of cytokines would not help. Perhaps this needs to be used in conjunction with a pathogen-focused test (ideally one with antimicrobial sensitivity testing baked in) for a truly holistic test.

We have expanded our discussion of this important limitation in paragraph 5 of the discussion. We believe that cytokine-based POC diagnostic tests will serve as important triage to rule out patients with ASB or non-infective conditions that may present with a similar symptomology, or to more easily identify true symptomatic UTI in patients who

cannot communicate symptoms effectively. By triaging these individuals a cytokine-based POC test could greatly reduce the empiric use of antibiotics. However, AST will still be critical to determine the best choice of antibiotics patients who are identified as true UTI positives.

Reviewer #2

Comments to the Authors (Required):

This manuscript by Ebrahimzadeh et al. investigates an important clinical issue, recurrent urinary tract infection. Urinary cytokines were measured in postmenopausal women with either rUTI or no recent UTI history with the goal of identifying a cytokine profile that could predict rUTI status and be developed as a point-of-care assessment. The study is a follow-up to a previous publication from the same group (Ebrahimzadeh et al. 2021), where PGE2 was found to be a marker that both signaled current rUTI and was predictive of future episodes. The data provided in the current manuscript provide a more refined diagnostic for rUTI over the previous manuscript. Although the longitudinal follow-up of the patient cohort is a strength, it was disappointing that the identified markers were not predictive of outcomes. The manuscript is well written, and the conclusions were straightforward.

I have one major concern: prior work on the identified cytokines in human populations needs to be more prominently stated to properly establish the context of this study. Specifically, IL-8 was found to be elevated in patient UTI urine samples at least as early as 1993 (PMID: 8454332, not cited), and has been previously proposed as a UTI diagnostic marker, although this cytokine may be elevated for other reasons (PMID: 11517116, also not cited). Thus, the advances in the current study may be more specifically applied to rUTI in postmenopausal women (an understudied cohort), and the major strength is the combinatorial panel of cytokines that was identified as a useful diagnostic. The inclusion of a marker that is elevated in the non-UTI "Never" group, IL-13, is especially interesting.

Thank you for pointing out these references we overlooked. We have added them to both the introduction and the discussion. We also found it very interesting that Ko et. al compared urine and serum IL-8 levels and found higher levels of IL-8 in urine supporting the idea of localized production. We included a sentence about this in the second paragraph of the discussion.

Minor comments follow.

1. Title: specify "postmenopausal" women

Done

2. Line 98: would it be more accurate to state the "never" group has no recent UTI history? Based on the Methods (lines 266-7), it appears that patients with recent UTI (1x in previous

year) were classified as "sporadic" and excluded from the study. It would follow that the "never" group would likely include women with prior UTI history of over 1 year ago.

The sporadic UTI exclusion criterion was used to exclude patients who had presented with UTI at the time of urine donation that did not have a history of recurrent UTI. The No UTI history group had no clinical history of UTI found in their available clinical records and as communicated by the patients. Patients with documented UTI over a year ago would be excluded from the No UTI history (Never) group.

3. Line 121/Fig 2C: it would be helpful to include absolute numbers in Fig. 2C along with the percentages. For example, it would be clear that 3.2% is n=1.

At the request of another reviewer we have included a new supplemental table (Table S1) with the complete clinical urine culture data for the cohort. The readers can now refer to this table to see how many patients presented with a particular pathogen.

4. Line 143: specify the limit of detection, either here or in the Methods.

We have specified the LoD for each analyte in the methods section.

5. Line 197: typo "postmenopausal"

Fixed

6. Lines 233-6: please elaborate on why this finding is interesting. Only one of the cytokines from this study mentioned in the previous sentence was elevated in Rodhe et al. Were all the same cytokines tested in both studies? Was something unexpected? Should we conclude or expect that some of the mentioned cytokines will be more indicative of uncomplicated cystitis vs. rUTI?

We altered this statement and have added more discussion to put our findings in the context of previous findings in the field in the second and fourth paragraphs of the discussion section. Please see our reply to Reviewer 1's point 14, but we think that the optimal diagnostic urinary cytokines may be different between different demographic and clinical groups. This, along with differences in study design may describe some of the differences in cytokines identified between studies.

7. Line 253: please clarify the statement about LPS. First, LPS would be generically present for Gram-negative bacteria and not at all for Gram-positive bacteria or non-bacterial UTI pathogens, regardless of UTI vs ASB. Second, I could not find mention of "lipopolysaccharide," "LPS," or "endotoxin" in the Stapleton et al. reference provided and it may have been mis-cited. I am aware of studies stating some LPS are more inflammatory than others (e.g., David Klumpp's work), although how this would be incorporated into a diagnostic kit is not clear from this discussion point.

Our apologies, you are correct that LPS was not used in the diagnostic platform evaluated in Stapleton et al. It was a different bacterial cell surface marker that was not well described. Thank you for catching this error, we have fixed our description of this work in the discussion.

8. Lines 280-4 Methods and Fig. 2C: please clarify if only monospecies infections were used for classifying bacteriology.

Figure 2C was made using the major uropathogen detected, but the new Table S1 contains the complete urine culture information. We have clarified this in the figure 2 legend.

9. Please check that the supplementary tables display correctly in the final version. On my review copy, many items were obscured where lines wrapped and were cut off or superimposed.

Thank you we will double check this.

Reviewer #3 (Comments to the Authors (Required)):

Ebrahimzadeh et al submit new results of pro-inflammatory cytokine profiling of urine specimens from postmenopausal women with and without history of recurrent UTI. Recurrent UTI is a major medical burden for this population, and current clinical diagnostic methods (urine culture) often take >24 hours to obtain results. In this work, the authors used a retrospective clinical cohort to identify subjects with documented rUTI (and UTI at the time of sampling) and subjects with no history of UTI and UTI negative at the time of sampling. Using multiplex cytokine arrays and follow-up targeted ELISA, the authors found that a multivariate model of IL-8, PGE2 (measured in a prior study), and IL-13 displayed the highest specificity and sensitivity for rUTI prediction. This is an important finding for designing improved diagnostics for rUTI in postmenopausal patients. Overall, the study is well-designed, data is analyzed and interpreted appropriately, and the manuscript is well-written. One concern is the lack of clarity on whether all urine samples were subjected to urine culture or extended urine culture techniques. Additional specific comments are focused on areas that may benefit from additional discussion or clarification of the presented work.

Specific comments:

1. Lines 98-99: Unclear why an additional arm - spontaneous UTI (not rUTI) was not included in analyses? Although the ability to identify rUTI biomarkers in postmenopausal women is of clinical value, the ability to also detect UTI in this population would also be valuable.

This would be useful for a future study but for this study we chose to focus on postmenopausal women with rUTI as they are an understudied demographic. We will definitely include spontaneous UTI cohorts in our future evaluation of diagnostic cytokines.

2. Lines 101-103: Given the differences in age and BMI between cohort arms, were age and BMI evaluated as variables in cytokine analyses?

Yes, but they were not shown to contribute significantly to the diagnostic model. We have added the analysis including age and BMI in Figure S5. We found that the identified cytokines were the most important for model accuracy as removal of both age and BMI from the multivariate model results in a less than 5% decrease in mean accuracy. We have also included a brief description of these data in the results section.

3. Lines 98-99, lines 265-269, and lines 280-281: It is unclear whether all samples were cultured as a part of urine collection. Line 280 states that UC was performed on Relapse rUTI patients. Was UC/extended UC performed on never UTI patients to account for ASB? Urine culture (and perhaps dipstick) should be performed on samples from patients in the Never group to confirm absence of urinary bacteria. Patient reported symptoms would not delineate between patients with negative bacterial culture and patients with asymptomatic bacteriuria. Further, comparing the specificity of urine dipstick to urinary cytokines in this cohort could further enhance the predictive power of this model.

For this cohort, clinical urine culture was not performed on the never UTI patients. Only urine from relapse patients was cultured as per standard clinical protocols and urine from symptomatic relapse patients that fell below the clinical lab cutoff of 10^5 CFU/mL was cultured on Chromagar. Future work should certainly include an ASB group to determine the accuracy of our model in differentiating UTI and ASB, but UC and UA was not performed on the Never UTIs as they did not report symptoms. Also, we agree that direct comparison of our diagnostic model to UA dipstick would be extremely valuable in a follow-up study.

4. Figure S2 - consider performing hierarchical clustering to identify co-occurring cytokines. Also, please provide details on how z scores calculated. Do cytokine patterns correlate with pathogens cultured from urine? Do patients with UPEC have altered cytokine profiles compared to non-UPEC cases?

We have added a description of how Z-scores were calculated to the statistical analysis section of the methods. Hierarchical clustering analysis identified 6 cytokine clusters which are now reported in Figure S3A. We also performed correlation analysis to validate the clusters (co-occurring cytokines) and found strong correlations especially between the pro-inflammatory cytokines in the third cluster. These data are presented in new Figure S3B. We did not find any correlation between specific pathogens and urinary cytokines. Please see added Supplementary Figure S4.

5. Table 2/Figure 3/Figure S3 - Indicate limits of detection for each cytokine. Samples following below the limit of detection should be displayed below the limit of detection (indicate n of these samples). Samples falling below the limit of detection should not be carried into normalized values since nothing was detected.

We have specified the LoD for each cytokine in the methods section. For the multiplex cytokine assay, falling below the LoD is actually more like falling below the limit of accurate quantitation as signal was detected but it was not high enough to be accurately quantified. Anyway, we completely understand your point and recognize this is an issue when considering the normalized data for IL-4. We have recognized this limitation in the results section and why we focus our model on the un-normalized data.

6. Line 210: Has a multivariate model been applied here to determine if the pairing PGE2 and IL-8 increases the predictive power for recurrence?

In response to this query, we applied a COX proportional hazards model to compare PGE2 alone versus PGE2+IL-8 and found that adding IL-8 has decreased prediction power. We also performed Kaplan-Meier analysis with both PGE2 and IL-8 as variables and it was not significant.

7. Lines 233-236: a limitation of this study is that only rUTI and never UTI samples were compared. Extending these analyses to acute cystitis and ASB samples would be important future work - especially given the elevated rate of ASB in elderly, post-menopausal women.

We agree that it will be important to extend this analysis into acute cystitis and ASB cohorts in future work. We have added this important point to the fourth paragraph of the discussion.

January 19, 2024

RE: Life Science Alliance Manuscript #LSA-2023-02323R

Dr. Nicole J De Nisco
The University of Texas at Dallas
Biological Sciences
800 W. Campbell Road
BSB12.515
Richardson, Texas 75080

Dear Dr. De Nisco,

Thank you for submitting your revised manuscript entitled "Inflammatory markers for improved recurrent UTI diagnosis in postmenopausal women". We would be happy to publish your paper in Life Science Alliance pending final revisions necessary to meet our formatting guidelines.

- please address Reviewer 3's remaining comment
- please be sure that the authorship listing and order is correct
- please use the [10 author names et al.] format in your references (i.e., limit the author names to the first 10)
- please upload all figure files as individual ones, including the supplementary figure files; all figure legends should only appear in the main manuscript file
- please add callouts for Figures S4A, B and S6A, B to your main manuscript text

A. FINAL FILES:

B. MANUSCRIPT ORGANIZATION AND FORMATTING:

Sincerely,

Reviewer #1 (Comments to the Authors (Required)):

The authors have done a great job in responding to my points as well as the of the other referees - thank you. I appreciate that so much new data were added. I have no further comments and think this should be accepted.

Reviewer #3 (Comments to the Authors (Required)):

The authors have adequately addressed my concerns in the revised manuscript. My only additional comment is that, since the authors have clarified that urine culture was not performed on the "never UTI" group, that the phrase "UC negative" must be removed from Figure 1 (blue text box) since urine culture was not performed on these samples.

January 31, 2024

RE: Life Science Alliance Manuscript #LSA-2023-02323RR

Dr. Nicole J De Nisco
The University of Texas at Dallas
Biological Sciences
800 W. Campbell Road
BSB12.515
Richardson, Texas 75080

Dear Dr. De Nisco,

Thank you for submitting your Research Article entitled "Inflammatory markers for improved recurrent UTI diagnosis in postmenopausal women". It is a pleasure to let you know that your manuscript is now accepted for publication in Life Science Alliance. Congratulations on this interesting work.

DISTRIBUTION OF MATERIALS:

Again, congratulations on a very nice paper. I hope you found the review process to be constructive and are pleased with how the manuscript was handled editorially. We look forward to future exciting submissions from your lab.

Sincerely,
